# The Effectiveness of Mobile Cloud 12-Lead Electrocardiogram Transmission System in Patients with ST-Segment Elevation Myocardial Infarction

**DOI:** 10.3390/medicina58020247

**Published:** 2022-02-06

**Authors:** Toyonori Arinaga, Yasunori Suematsu, Ayumi Nakamura, Tomoki Imaizumi, Yohsuke Hanaoka, Toshimitsu Takagi, Hidenobu Koga, Hironori Tanaka, Yasuhiko Shokyu, Shin-ichiro Miura

**Affiliations:** 1Department of Cardiology, Fukuoka University School of Medicine, Fukuoka 814-0180, Japan; sakikawa293@gmail.com (T.A.); ysuematsu@fukuoka-u.ac.jp (Y.S.); 2Department of Cardiology, Shin-Yukuhashi Hospital, Fukuoka 814-0180, Japan; nakamura198221@yahoo.co.jp (A.N.); whitewillson32@yahoo.co.jp (T.I.); yohsuke831@yahoo.co.jp (Y.H.); ttaka.cir@hotmail.co.jp (T.T.); kogahide919@gmail.com (H.K.); 3Department of Emergency and Critical Care Medicine, Shin-Yukuhashi Hospital, Fukuoka 824-0026, Japan; hero1031dar11@yahoo.co.jp (H.T.); shokyu@shinyukuhashihospital.or.jp (Y.S.); 4Department of Cardiology, Fukuoka University Nishijin Hospital, Fukuoka 814-8522, Japan

**Keywords:** mobile cloud-based 12-lead electrocardiogram transmission system, emergency medical service, ST-segment elevation myocardial infarction, early reperfusion therapy, hospital stay

## Abstract

*Background**and Objectives*: Delay of reperfusion therapy is related to high mortality in cases of ST-segment elevation myocardial infarction (STEMI). Guidelines emphasize that the first-medical-contact-to-balloon (FMCTB) time should be within 90 min. A mobile cloud-based 12-lead electrocardiogram (MC-ECG) transmission system might be useful in such cases, especially in rural areas. *Materials and Methods*: From April 2019 to June 2021, both an MC-ECG transmission system and the conventional method in which a physician checks the ECG in a hospital (Conventional) were used for transport by emergency medical services in Shin-Yukuhashi Hospital, Fukuoka, Japan. During this period, 8684 consecutive patients were transported to this hospital. Among them, we investigated 48 STEMI patients. The MC-ECG group (*n* = 23) and the Conventional group (*n* = 25) were enrolled. *Results*: There was no significant difference in FMCTB time between the MC-ECG and Conventional groups (MC-ECG: 72.0 (60.5–107) min vs. Conventional: 80.0 (63.0–92.0) min, *p* = 0.77). The length of hospital stay in the MC-ECG group was significantly shorter than that in the Conventional group (12.0 (10.0–15.0) days vs. 16.0 (12.0–19.0) days, *p* = 0.039). The logistic regression model showed that patients’ non-use of MC-ECG was associated with a risk of more than 15-day length of hospital stay with an adjusted odd ratio of 0.08 (95% CI: 0.013–0.55, *p* = 0.0098). *Conclusions*: Using the MC-ECG, the length of hospital stay in patients with STEMI was significantly reduced.

## 1. Introduction

Myocardial infarction is still the leading cause of sudden death in Japan. Kitamura et al. reported that 56.1% of out-of-hospital deaths in Japan were of cardiac origin [1]. ST-segment elevation myocardial infarction (STEMI) patients will suffer the risks of later adverse cardiac events, even after they survived. Sawano et al. analyzed a Japanese nationwide cohort and reported that the overall 30-day all-cause mortality rate was 3.0% and the overall 1-year incidence of all-cause death was 7.1%. This report indicates the necessity of improving care for STEMI [2].

For STEMI patients, the guidelines of the Japanese Circulation Society [3], American Heart Association [4], and European Society of Cardiology [5] recommend that primary percutaneous coronary intervention (pPCI) as revascularization therapy should be performed within 90 min from the arrival of the emergency medical service (EMS). In other words, these guidelines recommend that the first-medical-contact-to-balloon (FMCTB) time should be less than 90 min. To improve clinical outcomes in patients with STEMI, prehospital care is important, as with other medical therapy. The national average transport time in Japan is 39.5 min [6]. Therefore, when a patient arrives at a hospital, there are only about 50 min left to achieve revascularization within 90 min. The Keichiku area is located in eastern Fukuoka, Japan, and has a population of around 183,300 [7,8]. There are only 2 pPCI-capable hospitals in this area. While most patients transported to Shin-Yukuhashi Hospital come from less than 20 km (12.4 miles) away, some are transported from up to 45 km (28 miles) away. Therefore, patients who live in such a rural area have a handicap in time-to-treatment.

Previous studies have shown that prehospital 12-lead electrocardiogram (ECG) reduced the relative risk of 30-day mortality in patients with STEMI [9,10]. However, with this system, the ECG must be checked at the hospital. A mobile cloud-based 12-lead electrocardiography (MC-ECG) system can upload 12-lead ECG to the cloud from an ambulance in the field via Bluetooth and a 3G network [11,12,13]. By using MC-ECG, a medical doctor such as an on-call cardiologist was able to check the 12-lead ECG from the ambulance while at his/her home, on a tablet. The cardiologist could then immediately convene an emergency catheter team from outside the hospital. The ECG in the cloud is encrypted and personal information is protected. In light of these observations, we investigated whether MC-ECG shortened FMCTB time in patients with STEMI who were transported to Shin-Yukuhashi Hospital.

## 2. Materials and Methods

### 2.1. Study Design

The protocol was approved by the Independent Review Board of Fukuoka University and Shin-Yukuhashi Hospital (U20-06-012). The MC-ECG system (SCUNA, Mehergen Group Holdings, Fukuoka, Japan) was introduced sequentially in the Keichiku area from April 2019 to June 2021. During this period, EMS in the Keichiku area used both an MC-ECG system and a conventional system in which a medical doctor checks the ECG at the hospital. We retrospectively collected data for 8684 consecutive emergency cases at Shin-Yukuhashi Hospital from April 2019 to June 2021. The patient selection criteria are shown in Figure 1. We divided these cases into patients who were managed with an MC-ECG system (MC-ECG group, *n* = 584) and those who were managed with a conventional system (Conventional group, *n* = 8100). We excluded patients who suffered from non-acute coronary syndrome (ACS) and non-STEMI, those who had already been diagnosed with STEMI by other medical institutions, those who had undergone pacemaker implantation, those who suffered from vasospastic angina, those who had history of coronary artery bypass graft surgery, those who had cardiac arrest before reperfusion, those who had received intubation or extracorporeal membrane oxygenation before reperfusion, those who did not undergo pPCI, those who had no available data, those with serious infectious disease, and those who had failure of transmission electrocardiogram system. STEMI was defined as biomarker evidence of myocardial injury with rise and/or fall of cardiac troponin value above the 99th percentile upper reference limit, with chest discomfort or other ischemic symptoms and ECG change including new ST elevation at the J-point in two contiguous leads with the cut-point: ≥1 mm in all leads other than leads V2–V3 where the following cut-points apply: ≥2 mm in men ≥40 years; ≥2.5 mm in men <40 years, or ≥1.5 mm in women regardless of age, increased hyperacute T wave amplitude with prominent symmetrical T waves in at least two contiguous leads, and/or new bundle branch blocks with ischemic repolarization patterns and/or development of pathologic Q waves on ECG [14]. Finally, 48 STEMI patients were transported to Shin-Yukuhashi Hospital and subjected to pPCI. We compared the 23 patients in the MC-ECG group to the 25 patients in the Conventional group.

### 2.2. Baseline Characteristics

Age, gender, body mass index, current or past smoker, hypertension, diabetes mellitus, dyslipidemia, chronic kidney disease, and history of ischemic heart disease were investigated. Body mass index (BMI) was calculated as body weight (kg)/height (m)^2^ and BMI ≥ 25 kg/m^2^ patients were defined as obese. The transport distance and the rate of off-hour arrival were also investigated, where off-hour was defined as 6:00 p.m.–7:00 a.m.

### 2.3. Clinical Pathology and Outcomes of STEMI

Killip classification, culprit coronary artery, number of diseased vessels, peak creatine phosphokinase (CK), peak CK-MB, left ventricular ejection fraction, brain natriuretic peptide (BNP), 30-day mortality, length of stay in the intensive care unit (ICU), and length of hospital stay were investigated.

### 2.4. Transport Time

Collapse-to-first-medical-contact time, collapse-to-balloon time, FMCTB time, door-to-catheter lab room time, and door-to-balloon time were investigated. We defined first medical contact as the time the ambulance arrived at the scene. Balloon time was defined as when the first balloon was inflated or when a thrombus aspiration catheter was deployed.

### 2.5. Statistical Analyses

All data analyses were performed using EZR (Saitama Medical Center, Jichi Medical University, Saitama, Japan), which is a graphical user interface for R (The R Foundation for Statistical Computing, Vienna, Austria) [15]. More precisely, it is a modified version of R commander designed to add statistical functions frequently used in biostatistics. Continuous variables with a normal distribution are expressed as mean ± standard deviation and compared between the groups by Student’s t-test. Continuous variables with a non-normal distribution are expressed as a median (interquartile range) and compared between the groups by the Mann–Whitney U test. Differences were evaluated by a two-sided test with an alpha level of 0.05. Categorical variables were compared between the groups by the chi-square test, and variables with expected values less than 5 were compared by Fisher’s exact test. When we convert continuous variables into categorical variables, we draw receiver operating characteristic (ROC) curves to set appropriate cut-offs. We used multivariable logistic regression to control for the potentially confounding roles of five variables: age, sex, the use of MC-ECG, and two variables which were selected by backward stepwise method. If these continuous variables were with a non-normal distribution, they were logarithmically converted or were squared. A value of *p* < 0.05 was considered significant.

## 3. Results

### 3.1. Patient Characteristics at Baseline

Table 1 shows the patient characteristics at baseline. In all patients (*n* = 48), age, percentage of males, and BMI were 77.5 (67.3–81.0) years, 72.9% (*n* = 35), 23.5 (21.5–26.1) kg/m^2^, respectively. The percentage of current and past smokers in the MC-ECG group was significantly higher than that in the Conventional group (*p* = 0.045). There were no differences in the percentages of hypertension, diabetes mellitus, dyslipidemia, chronic kidney disease, and the history of ischemic heart disease between the MC-ECG and Conventional groups. In all patients, transport distance and the percentage of off-hour arrival were 8.5 (5.6–16.1) km and 43.8% (*n* = 21), respectively, and there were no significant differences in these factors between the groups.

### 3.2. Clinical Pathology and Outcomes

Table 2 shows the clinical pathology of STEMI. Among all patients, 37.5% of patients were Killip I (*n* = 18) and 25.0% were Killip IV (*n* = 12). Among all patients, 91.7% of patients got TIMI 3 flow grade after procedure. The most common culprit coronary artery was the left anterior descending artery (45.8%, *n* = 22) and 75.0% of patients had multivessel disease (*n* = 36), again among all patients. There were no significant differences in clinical pathology of STEMI between the MC-ECG and Conventional groups.

Table 3 shows the clinical outcomes. In all patients, peak CK and CK-MB were 958 (532–2329) IU/L and 85 (44–299) IU/L, respectively. The 30-day mortality was 2.1% (*n* = 1), and the patient, who was in the MC-ECG group, died due to cerebral embolism. Length of stay in the ICU was 3.0 (2.0–3.3) days in all patients. There were no significant differences in the 30-day mortality and length of stay in the ICU between the MC-ECG and Conventional groups. On the other hand, length of hospital stay in the MC-ECG group was significantly shorter than that in the Conventional group (*p* = 0.04). 

### 3.3. Transport Time

As shown in Figure 2, there were no differences in collapse-to-balloon time (138 (92–182) min vs. 105 (96 –161) min, *p* = 0.66), collapse-to-first-medical-contact time (42.0 (21.5–74.0) min vs. 35.0 (14.0–72.0) min, *p* = 0.56), FMCTB time (72.0 (60.5–107.0) min vs. 80.0 (63.0–92.0) min, *p* = 0.77), door-to-balloon time (49.0 (41.0–85.5) min vs. 59.0 (38.0–67.0) min, *p* = 0.63), and door-to-catheter laboratory time (24.0 (17.5–44.0) min vs. 24.0 (15.0–41.0) min, *p* = 0.73) between the MC-ECG and Conventional groups.

### 3.4. Multivariable Logistic Regression Analysis to Predict 15-Day or More Hospital Stay

The use of MC-ECG may be associated with length of hospital stay, as shown in Table 3. As shown in Figure 3, a ROC curve analysis showed that the area under the curve for length of hospital stay were 0.68 in all patients. The cut-off level of length of hospital stay that gave the greatest sensitivity and specificity for the use of MC-ECG system was 15.5 days (sensitivity 0.83, specificity 0.56). We divided the patients into two groups according to length of hospital stay: equal or less than 15 days and more than 15 days, and then performed multivariable logistic regression analysis. For the analysis, diabetes mellitus and peak logarithmically converted CK levels as independent variables were selected by the stepwise method. Body mass index, current or past smoker, hypertension, dyslipidemia, chronic kidney disease, history of ischemic heart disease, transport distance, the rate of off-hour arrival, Killip classification, the number of diseased vessels, the use of intra-aortic balloon pumping, complication of heart failure after admission, complication of atrial fibrillation, collapse-to-balloon time, and FMCTB time as variables were eliminated by a step-wise method. 

Figure 4 shows the odd ratios (OR) and 95% confidence intervals (CI) from logistic regressions of more than 15-day length of hospital stay that adjust for the patient and setting characteristics as described previously. Patients with diabetes mellitus and logarithmically converted high levels of peak CK were associated with a high risk of more than 15-day length of hospital stay, with an adjusted OR of 8.8 (95% CI: 1.3–59.4, *p* = 0.027) and 19.2 (95% CI: 2.4–151.0, *p* = 0.005), respectively. Patients with the use of MC-ECG were associated with a reduced risk of more than 15-day length of hospital stay with an adjusted OR of 0.08 (95% CI: 0.01–0.55, *p* = 0.010). Other factors, such as age and gender, were not associated with a risk of over 15-day length of hospital stay. 

## 4. Discussion

The main finding in this study was that length of hospital stay in the MC-ECG group was significantly shorter than that in the Conventional group.

The length of hospital stay in the MC-ECG group was significantly shorter than that in the Conventional group, although there were no significant differences in clinical outcomes including peak CK, peak CK-MB, 30-day mortality, length of stay in ICU between the groups, and the ratio of smoking in the MC-ECG group was higher than that in the Conventional group. A possible explanation is that introduction of MC-ECG had improved the EMS team’s management ability in use of 12-leads electrocardiogram. When an EMS team suspected ACS and used the MC-ECG system, it could have led to bidirectional communication between an EMS team and physicians through that procedure. As a result, it was likely to have reduced the likelihood of transporting an ACS patient to a facility that is not capable of pPCI or has no board-certified cardiologists [16], and could have encouraged appropriate prehospital care including appropriate oxygen administration, avoiding routine administration for the patients [3]. Several reports showed that early treatment in some conditions has a positive effect on clinical outcomes. Specifically, AMI patients were treated by cardiologists directly or under consultation between cardiologists and general physicians [17,18,19,20]. The MC-ECG enabled a cardiologist belonging to this hospital to immediately check and diagnose all patients in the MC-ECG group. Under these conditions, MC-ECG may have somewhat improved clinical outcomes. The EMS team communicates with a hospital by their mobile phone. There are only two hospitals that are able to treat AMI patients in the Keichiku area. According to the 2020 census of Japan, the population density of whole Fukuoka prefecture was 1029.8 per km^2^ and that of the Keichiku area was 322.10 per km^2^. Reduced hospitalization is a benefit for the whole population because of the fast turnover of patients. SCUNA should be more useful for rural hospitals, because sometimes a cardiologist does not remain onsite at a rural hospital during off-hours. This merit was apparent in this study because the rate of off-hour arrival was high (43.8%). If the ratio of smoking in the MC-ECG group was similar to that of the Conventional group, a more favorable result might be shown. 

High serum levels of peak CK, the presence of diabetes mellitus, and no use of MC-ECG were associated with more than 15-day hospital stay (Figure 4). Patients with low serum levels of peak CK may have recovered from severe clinical condition smoothly and rehabilitated with a low risk of adverse events [21]. Patients without diabetes mellitus, which is a strong risk factor for coronary artery disease mortality [22], may have been discharged from the hospital in a shorter period of time, because they did not need additional coordination of oral hypoglycemic drug, insulin, nor patient education. On the other hand, obesity, defined as BMI ≥ 25 kg/m^2^, was not related with diabetes or other clinical outcomes [23]. As shown in Appendix A, the differences of length of hospital stay were not associated with complications of heart failure. In addition, the length of hospital stay was not associated with transport time including collapse-to-balloon time, FMCTB time, door-to-balloon time, or door-to-catheter lab time.

This study cannot demonstrate that MC-ECG was effective for shortening the time from first medical contact-to-reperfusion in patients with STEMI. Our sample size was small for evaluating the effectiveness of MC-ECG about FMCTB and door-to-balloon times. Prehospital 12-lead ECG improved the transport time and 30-day mortality for both STEMI and non-STEMI patients [24]. Prehospital tele-ECG shortened the door-to-balloon time and reduced in-hospital mortality in rural acute coronary syndrome patients [25]. Nowadays, machine learning-based prediction of prehospital 12-lead ECG has been reported [26]. In the future, EMS might be able to diagnose the severity of acute coronary syndrome using these systems. MC-ECG was advocated in 2013 [27]. SCUNA is a MC-ECG transmission system [11,12,13]. SCUNA was shown to be effective for reducing door-to-balloon time in patients with acute coronary syndrome [28]. Previous studies, such as Kawakami et al., had reported in 2016 [29] that FMCTB time was 97 (82–112) min in the group with the use of prehospital electrocardiogram. However, we reported FMCTB time was 80 (63–92) min in this study, even in the Conventional group. Similarly, door-to-balloon time and door-to-catheter laboratory time of prehospital-ECG group in the previous study were 68 (52–83) min and 27 (20–36) min. However, those of the Conventional group in this study were 59 (38–67) min and 24 (15–41) min, espectively. There was not a major difference of achieved time, comparing these studies, even though the previous one was an intervention group and this study was a Conventional group. Previous studies, such as Kojima et al., had reported on Japanese nationwide database of acute myocardial infarction in 2018 [30] that the median door-to-balloon time was 81 (53–143) min and collapse-to-balloon time was 230 (141–420) min; this was an analysis of overall AMI including 79.7% STEMI. The report was based on Japanese nationwide real-world data of AMI. However, we reported that door-to-balloon time and collapse-to-balloon time were 59 (38–67) min and 138 (92–182) min in the Conventional group, respectively. Although it is not possible to simply compare a previous study and this study, the time to reperfusion tended to be shorter in this study. It may be suggested that Shin-Yukuhashi Hospital had been equipped with a cardiovascular emergency system, and that the facility had kept a good-working relationship with EMS from before introduction of the MC-ECG system. These could have been reasons for there to be no significant differences in collapse-to-balloon time, FMCTB time, door-to-balloon time, and door-to-catheter lab time, respectively, by using the MC-ECG system. 

This study had several limitations. First, this was a single-center, retrospective observational study, and thus the number of patients might be insufficient. However, we have demonstrated a part of effectiveness of MC-ECG despite the fact that it is difficult to recruit STEMI patients in such a rural area. Second, we were unable to indicate positive impact of MC-ECG on transport time. This result does not correspond to evidence from previous studies, where the reperfusion timing was strongly associated with clinical outcomes [31,32,33,34]. Small sample size must be an influence here. MC-ECG was introduced in the Keichiku area, Fukuoka, Japan from April 2019. There is a possibility that EMS was not used to the system at first. 

## 5. Conclusions

We showed that 15-day or less length of hospital stay was significantly associated with the use of an MC-ECG transmission system.

## Figures and Tables

**Figure 1 medicina-58-00247-f001:**
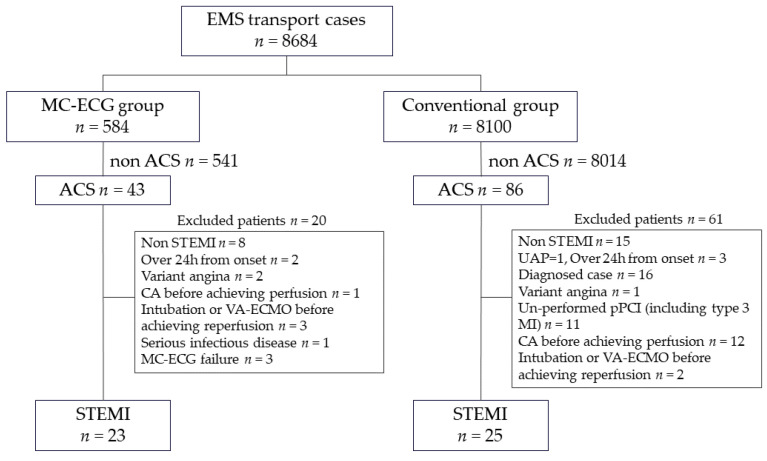
Patient selection criteria. EMS: emergency medical service, MC-ECG: mobile cloud 12-lead electrocardiogram, ACS: acute coronary syndrome, UAP: unstable angina pectoris, Type3 MI: Type3 myocardial infarction, STEMI: ST-segment elevation myocardial infarction, pPCI: primary percutaneous coronary intervention, CA: cardiac arrest, VA-ECMO: veno-arterial extracorporeal membrane oxygenation. Type 3 MI is defined as patients who suffer cardiac death, with symptoms suggestive of myocardial ischemia accompanied by presumed new ischemic electrocardiogram changes or ventricular fibrillation, but who die before blood samples for biomarkers can be obtained, or before increases in cardiac biomarkers can be identified, or MI is detected by autopsy examination.

**Figure 2 medicina-58-00247-f002:**
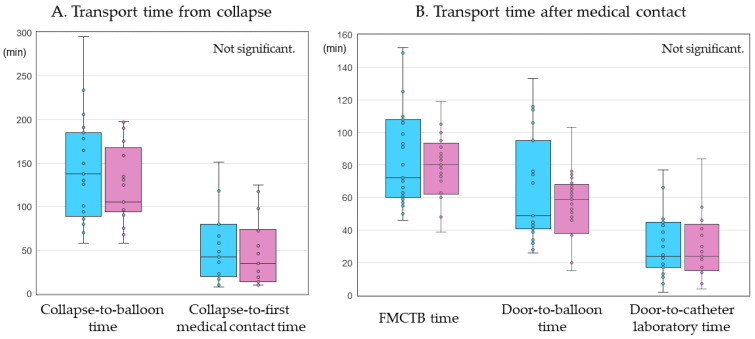
Transport time. (**A**) Transport time from collapse at collapse-to-balloon time and collapse-to-first medical contact time. (**B**) Transport time after first medical contact at first medical contact-to-balloon (FMCTB) time, door-to-balloon time, and door-to-catheterization laboratory time. Blue and red bars indicate the MC-ECG group and Conventional group, respectively. MC-ECG: mobile cloud-based 12-lead electrocardiogram.

**Figure 3 medicina-58-00247-f003:**
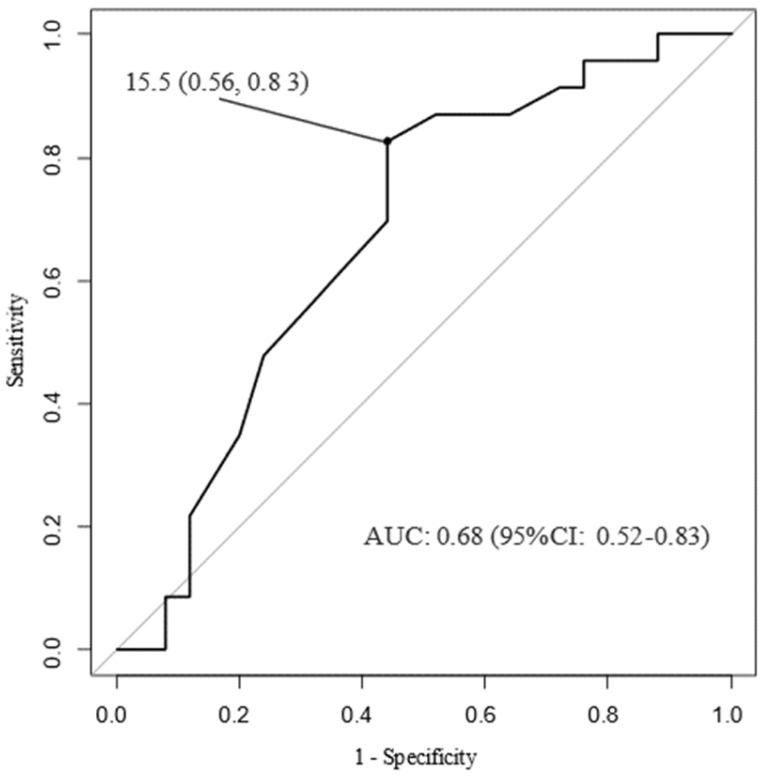
Receiver operating characteristic curves. ROC: receiver operating characteristic, CI: confidence interval, AUC: area under the curve.

**Figure 4 medicina-58-00247-f004:**
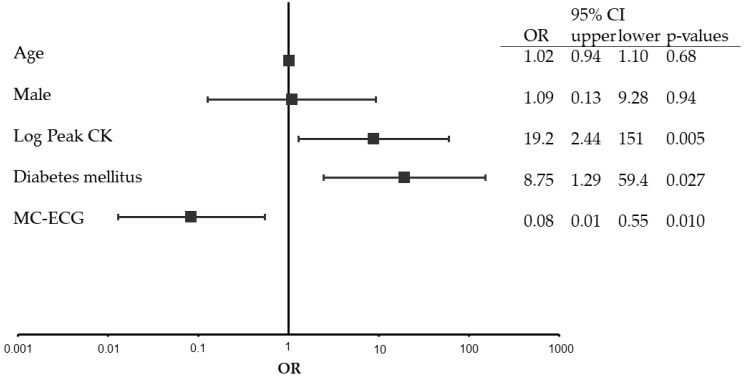
Multivariable logistic regression. CI: confidence interval, OR: odds ratio, CK: creatine phosphokinase, MC-ECG: mobile cloud-based 12-lead electrocardiogram.

**Table 1 medicina-58-00247-t001:** Patient characteristics.

Variables	MC-ECG Group	Conventional Group	*p*-Value
	(*n* = 23)	(*n* = 25)	
Age, years	76.0 (61.5–80.5)	78.0 (71.0–82.0)	0.4
Male, *n* (%)	16 (69.6)	19 (76.0)	0.8
Body mass index, kg/m^2^	24.6 ± 4.0	23.3 ± 4.0	0.3
Current and past smoker, *n* (%)	17 (73.9)	11 (44.0)	0.045
Hypertension, *n* (%)	15 (65.2)	20 (80.0)	0.3
Atrial fibrillation, *n* (%)	2 (8.7)	2 (8.0)	1.0
Diabetes mellitus, *n* (%)	6 (26.1)	10 (40.0)	0.4
Dyslipidemia, *n* (%)	12 (52.2)	12 (48.0)	1.0
Chronic kidney disease, *n* (%)	5 (21.7)	7 (28.0)	0.7
Hemodialysis, *n* (%)	0 (0)	1 (4.0)	1.0
Past history of IHD, *n* (%)	6 (26.1)	5 (20.0)	0.7
Left ventricular ejection fraction, (%)	51.7 ± 11.9	51.7 ± 11.6	1.0
BNP, pg/mL	53 (24–125)	68 (27–229)	0.6
Transport distance, km	7.1 (5.8–14.2)	9.4 (5.3–19.0)	0.7
Arrival in off hour, *n* (%)	11 (47.8)	10 (40.0)	0.8

MC-ECG: mobile cloud 12-lead electrocardiogram, IHD: ischemic heart disease, BNP: brain natriuretic peptide, off-hour: defined as 6:00 p.m.–7:00 a.m.

**Table 2 medicina-58-00247-t002:** Clinical pathology of STEMI.

Variables	MC-ECG Group	Conventional Group	*p*-Value
	(*n* = 23)	(*n* = 25)	
Killip classification, *n* (%)			0.2
I	7 (30.4)	11 (44.0)	
II	9 (39.1)	4 (16.0)	
III	3 (13.0)	2 (8.0)	
IV	4 (17.4)	8 (32.0)	
Culprit coronary artery, *n* (%)			0.3
Left main trunk	0 (0.0)	3 (12.0)	
Left anterior descending artery	13 (56.5)	9 (36.0)	
Left circumflex artery	2 (8.7)	2 (8.0)	
Right coronary artery	8 (34.8)	11 (44.0)	
Diseased Vessel, *n* (%)			0.4
Single	8 (34.8)	4 (16.0)	
Double	8 (34.8)	10 (40.0)	
Triple	7 (30.4)	11 (44.0)	
Post-procedural TIMI flow grade, *n*(%)			1.0
0	0 (0)	0 (0)	
1	0 (0)	1 (4.0)	
2	1 (4.3)	2 (8.0)	
3	22 (95.7)	22 (88.0)	
IABP, *n* (%)	2 (8.7)	2 (8.0)	1.0

STEMI: ST-segment elevation myocardial infarction, MC-ECG: mobile cloud 12-lead electrocardiogram, Diseased Vessel was defined as over 50% stenosis, IABP: intra-aortic balloon pumping.

**Table 3 medicina-58-00247-t003:** Clinical outcomes of STEMI.

Variables	MC-ECG Group	Conventional Group	*p*-Value
	(*n* = 23)	(*n* = 25)	
Peak CK, IU/L	1031 (583–2554)	956 (325–1725)	0.6
Peak CK-MB, IU/L	85 (49–305)	72 (40–207)	0.4
Complication of heart failure after PCI, *n* (%)	7 (30.4)	5 (20.0)	0.5
30-days mortality, *n* (%)	1 (4.3)	0 (0.0)	0.5
Length of stay in ICU, days	3 (2–3)	3 (2–4)	0.5
Length of hospital stay, days	12 (10–15)	16 (12–19)	0.04

STEMI: ST-segment elevation myocardial infarction, MC-ECG: mobile cloud 12-lead electrocardiogram, CK: creatine phosphokinase, ICU: intensive care unit.

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
