# Peer review of "The Effectiveness of Mobile Cloud 12-Lead Electrocardiogram Transmission System in Patients with ST-Segment Elevation Myocardial Infarction"

_medicina, 2022, doi:10.3390/medicina58020247_

Round 1

Reviewer 1 Report

Dear Author:

This is an interesting manuscript which stated the importance of Mobile Cloud ECG transport system. Several question need author to answer 

  1. The Mobile Cloud ECG transport system via Bluetooth may be the new system which acquired after 2020. The differences of medical resources may exist due to different period of time. Author should empasize these in their manuscipt
  2. It is strange that whenever the Mobile Cloud ECG transport system work, the FMCTB cannot reduced than conventional group. Maybe the patient number is too small to analysis. Author should clarify this in limitation
  3. The reason of why MC-ECG group can reduce should be emphasized more in discusssion. However, the differences of patient group should also be emphasized because the Smoker percentage is different
  4. The article by HY Fang, WC Lee, Medicine 2021;100:26(e26558) can be adopted and explain the differences between group. The length from patient to the hospital can be analysed if possible

Thank you very much !

Author Response

We would like to thank you for your valuable suggestions. We have addressed each point as shown below and hope that our responses are acceptable.

Comments and Suggestions for Authors

Dear Author: This is an interesting manuscript which stated the importance of Mobile Cloud ECG transport system. Several question need author to answer

  1. The Mobile Cloud ECG transport system via Bluetooth may be the new system which acquired after 2020. The differences of medical resources may exist due to different period of time. Author should empasize these in their manuscript.

Ans.) Thank you for your suggestion. The Mobile Cloud ECG transport system via Bluetooth was introduced in the Keichiku area, Fukuoka, Japan from April 2019. In our study period, between April 2019 and June 2020, the medical resources were almost unchanged. But there is a possibility that emergency medical service was not used to the system at first. We did not consider this limitation. We added this limitation in the limitation section.

  1. It is strange that whenever the Mobile Cloud ECG transport system work, the FMCTB cannot reduced than conventional group. Maybe the patient number is too small to analysis. Author should clarify this in limitation

Ans.) Thank you for your suggestion. We added this limitation in the limitation section.

  1. The reason of why MC-ECG group can reduce should be emphasized more in discusssion. However, the differences of patient group should also be emphasized because the Smoker percentage is different.

Ans.) Thank you for your suggestion. We moved the reason why MC-ECG group reduced hospital stay to an appropriate place and added the difference of smoking ratio in the discussion section.

  1. The article by HY Fang, WC Lee, Medicine 2021;100:26(e26558) can be adopted and explain the differences between group. The length from patient to the hospital can be analysed if possible

Ans.) Thank you for your suggestion. We showed the transport distance in Table 1, but there was no significant difference between the groups.

Reviewer 2 Report

The authors attempted to show the benefit of ECG transmission by mobile cloud system for STEMI patients in their retrospective study. They compared transport time after medical contact and hospital stay between MC-ECG group and conventional group. They concluded that the length of hospital stay in patients with STEMI was significantly reduced whereas FMCTB time was not. The positive result is regarded as reliable because it is significant even in the presence of small size. However, number of the patients is too small to conclude that FMCTB and DTB times were not significantly different. In the light of the limitation of size, the negative results regarding FMCTB and DTB times may be elusive. Thus this reviewer recommend that they revise this paper with more number of patients, or limit the conclusion to the positive results.  

Author Response

We would like to thank you for your important questions and valuable suggestions. We have addressed each point as shown below and hope that our responses are acceptable.

Comments and Suggestions for Authors

The authors attempted to show the benefit of ECG transmission by mobile cloud system for STEMI patients in their retrospective study. They compared transport time after medical contact and hospital stay between MC-ECG group and conventional group. They concluded that the length of hospital stay in patients with STEMI was significantly reduced whereas FMCTB time was not. The positive result is regarded as reliable because it is significant even in the presence of small size. However, number of the patients is too small to conclude that FMCTB and DTB times were not significantly different. In the light of the limitation of size, the negative results regarding FMCTB and DTB times may be elusive. Thus this reviewer recommend that they revise this paper with more number of patients, or limit the conclusion to the positive results. 

Ans.) Thank you for your suggestion. We agree with your suggestion. We deleted the negative results regarding FMCTB and DTB times from conclusions and added the limitation about the small sample size in the discussion section.